# Methods and Technologies for Supporting Knowledge Sharing within Learning Communities: A Systematic Literature Review

**Majid Zamiri [1,\*] and Ali Esmaeili [2]**

1. NOVA School of Science and Technology, Center of Technology and Systems (UNINOVA-CTS) and Associated Lab of Intelligent Systems (LASI), NOVA University Lisbon, 2829-516 Campus de Caparica, Portugal
2. Mechanical Engineering Department, Faculty of Engineering, Ferdowsi University of Mashhad, Mashhad 9177948974, Iran; aliesmaeili@ferdowsi.um.ac.ir
* Correspondence: ma.zamiri@campus.fct.unl.pt

**Abstract:** In an era marked by swift technological advancements and an escalating emphasis on collaborative learning, understanding effective methods and technologies for sharing knowledge is imperative to optimize educational outcomes. This study delves into the varied methods and technologies applied to facilitate and support knowledge sharing within learning communities. To achieve this, a systematic literature review was conducted, systematically collecting and scrutinizing pertinent literature. Employing automated searches, title-based selection, and reputation-based filters ensured the inclusion of high-quality studies. The chosen studies underwent a meticulous evaluation, considering factors like relevance, methodological robustness, and currency. The literature review unveiled a diverse array of methods and technologies employed in learning communities to facilitate effective knowledge sharing. The outcomes of this study offer a comprehensive snapshot of the existing literature, underscoring the significance of methods and technologies in supporting knowledge sharing within learning communities. By comprehending the strengths, challenges, and potential future trajectories, educators, researchers, and policymakers can make informed decisions to enhance the efficacy of knowledge sharing within learning communities.

**Keywords:** knowledge sharing; learning communities; methods; technologies; learners

## 1. Introduction

In the vast landscape of education, the concept of learning communities has emerged as a transformative force, transcending traditional classroom boundaries. At the heart of this evolution is the fundamental principle of knowledge sharing—a dynamic process that not only enhances individual learning but also cultivates a collaborative environment where insights flourish. The modern educational paradigm emphasizes collaborative learning, making it imperative to understand and employ effective methods and strategies for knowledge sharing (Abdul Rahman et al. 2014; Pappas et al. 2017).

Learning communities are collaborative, interactive, and often interdisciplinary spaces where individuals (learners) with common interests, goals, or educational pursuits come together to exchange their sharable knowledge, experiences, and resources. Learning communities can be founded at different levels, ranging from small to mass levels (Zamiri and Camarinha-Matos 2021). They can take various forms, adapting to different contexts and purposes. The main examples of learning communities across different settings include, but are not limited to, classroom-based learning communities, online forums and discussion groups, professional learning communities (PLCs), project-based collaborative teams, communities of practice (CoPs), residential learning communities, faculty learning communities, online courses and MOOCs, subject-specific communities, corporate learning communities, student organizations and clubs, and communities for lifelong learners (Rook et al. 2020; Otto et al. 2015; Zamiri 2022). These examples highlight the diverse ways in which learning

communities can be structured to facilitate collaboration, engagement, and shared learning experiences. The effectiveness of these communities depends on various factors, including the context, goals, active participants, facilitation, technology infrastructure, and alignment with the goals of knowledge sharing (West and Williams 2017). Furthermore, the choice of a learning community should align with the specific needs and objectives of the participants involved. For example, PLCs, in the context of education, play a pivotal role in fostering knowledge sharing among educators and professionals. These communities provide a structured and collaborative space where individuals with a shared purpose come together to enhance their practices. In PLCs, knowledge sharing is a cornerstone, facilitated by regular meetings, discussions, and collaborative projects. Educators within PLCs share insights, best practices, and resources, creating an appropriate environment for continuous learning (Admiraal et al. 2021). In another example, CoPs bring together learners with shared objectives, offering a setting for voluntary, informal, and self-directed collaborations. By promoting open communication, creative methodologies, and the exchange of experiences and expertise, these communities facilitate knowledge sharing, learning, and adaptation. With these attributes, CoPs are a dynamic learning ecosystem, propelling both individual and collective development (Matsuo and Aihara 2022).

The common goals associated with learning communities include, but are not limited to, promoting collaborative learning, building a sense of community, facilitating interdisciplinary perspectives, enhancing communication and critical thinking skills, supporting academic success, encouraging active participation, fostering personal growth and development, and preparing for real-world challenges. The specific goals of a learning community are often defined collectively by its members and organizers based on their shared interests and needs. It is essential to recognize that the goals of learning communities are not mutually exclusive and can evolve. The adaptability of these goals allows learning communities to cater to the dynamic needs and aspirations of their participants (Smith and MacGregor 2009; Virtue et al. 2019; Sai and Saedah 2015).

Learning communities exhibit several key characteristics that distinguish them from traditional educational settings. These characteristics contribute to creating environments that promote collaboration, engagement, and shared learning experiences. Here are some key characteristics of learning communities (Wise and Yi 2018; Schaap and Bruijn 2018; Zhang and Sun-Keung Pang 2016; Bielaczyc and Collins 1999):

- Shared Goals and Interests: learning communities typically revolve around a common purpose, whether it is a specific academic subject, a professional field, or a particular area of interest. Participants share common goals and objectives.
- Interactivity and Collaboration: interaction is a fundamental aspect of learning communities. Participants actively engage with each other, fostering a collaboration, discussion, and the exchange of insights. This can occur through face-to-face meetings, online discussions, or collaborative projects.
- Diversity of Participants: learning communities bring together learners with diverse backgrounds, experiences, and perspectives. This diversity enriches the learning environment by providing varied viewpoints and approaches to problem-solving.
- Supportive Environment: learning communities create a supportive environment where individuals feel comfortable expressing their ideas, asking questions, and seeking assistance. This environment encourages active participation and the development of a sense of belonging.
- Active Learning Strategies: active learning is often emphasized in learning communities. Participants can engage in activities, such as group projects, peer teaching, and hands-on experiences, to deepen their understanding of the subject matter.
- Integration of Formal and Informal Learning: learning communities often bridge the gap between formal and informal learning. While formal instruction can take place, informal learning through discussions, shared resources, and collaborative exploration is equally valued.

- Flexibility in Structure: learning communities can exist within various structures, ranging from formal educational institutions to online platforms, professional organizations, or workplace teams. The flexibility of these structures allows for the adaptation to the unique needs of the participants.

Learning communities contribute significantly to the development of well-rounded learners equipped with both knowledge and essential skills. Learning communities play a crucial role in education and professional development, offering a range of benefits that contribute to the overall growth and success of learners and the community (Corneli and Mikroyannidis 2012). Here are some key benefits of learning communities for learners and the community (Weiss et al. 2015; VanOra 2019; Arensdorf and Naylor-Tincknell 2016):

- Enhanced Learning Experience: learning communities provide an environment where learners can actively engage with the subject matter. Collaborative learning, discussions, and shared experiences contribute to a richer and more dynamic learning experience.
- Promotion of Active Learning: learning communities emphasize active learning strategies, such as group projects, case studies, and discussions. This fosters a deeper understanding of the material as learners actively apply concepts to real-world contexts.
- Diversity and Inclusivity: by bringing together learners with diverse backgrounds, experiences, and perspectives, learning communities promote diversity and inclusivity. This diversity enriches the learning environment, exposing learners to different viewpoints and approaches.
- Peer Support and Mentorship: learning communities create a support system where learners can seek help, share resources, and provide mentorship to one another. Peer support enhances the overall learning experience and contributes to personal and academic growth.
- Holistic Development: learning communities often focus on holistic development, emphasizing not only academic knowledge but also skills, such as critical thinking, communication, collaboration, and problem-solving.
- Community Building and Networking: learners in learning communities build strong connections with their peers, educators, and professionals. These networks can extend beyond the learning environment, providing valuable contacts for future collaborations and career opportunities.
- Interdisciplinary Learning: learning often encourages interdisciplinary approaches, breaking down traditional disciplinary boundaries. This approach promotes a broader understanding of subjects and fosters creativity for problem-solving.
- Cultivation of Lifelong Learning Habits: being a part of a learning community instills a mindset of continuous learning. Learners are more likely to adopt lifelong learning habits, staying curious and open to new ideas throughout their academic and professional journeys.
- Increased Engagement and Motivation: learning communities create an engaging and motivating atmosphere. Active participation, collaborative projects, and a sense of community inspire learners to be more invested in their learning.
- Real-World Relevance: many learning communities incorporate real-world applications of knowledge. This helps learners understand the practical implications of what they are learning, preparing them for challenges in their academic or professional pursuits.
- Support for Diverse Learning Styles: learning communities recognize and accommodate diverse learning styles. This flexibility ensures that learners with different preferences and strengths can thrive in a collaborative environment.
- Positive Impact on Retention Rates: the research suggests that learners (e.g., students and workers) who participate in learning communities often experience higher retention rates. A sense of belonging and support contribute to increased persistence and success.
- Professional Development Opportunities: in professional settings, participation in learning communities offers learners (e.g., employees) opportunities for continuous professional development, skill enhancement, and staying updated on industry trends.

In sum, learning communities contribute significantly to creating positive, supportive, and dynamic learning environments that foster academic, professional, and personal developments. Learning communities can also support educational institutions, businesses, and different sectors (government, public, or private), aiming to create meaningful and impactful learning experiences (Wilsona et al. 2023).

*1.1. Knowledge Sharing in Learning Communities*

In the dynamic environment of education, learning communities have emerged as powerful catalysts for collaborative and engaging learning experiences. At the heart of these communities lies a fundamental principle: the sharing of knowledge. Knowledge sharing refers to the process of exchanging ideas, insights, experiences, and expertise among individuals or groups. It involves the transmission of knowledge from one person to another, intending to increase the collective understanding and capabilities in a society, organization, or community. Knowledge sharing can take various forms, including, but not limited to, verbal communication, written documentation, interactive discussions, collaborative projects, and the use of digital platforms (Huysman and Wit 2002; Zamiri and Baqutayan 2012a). The following are the key aspects of knowledge sharing (Diab 2021; Zheng 2017; Farhan and Muhaimin 2019):

- Transfer of knowledge: knowledge sharing involves the communication of the understandings, findings, and experiences from one entity to another. This transfer can occur through various channels, such as conversations, presentations, documents, or multimedia.
- Collaboration and Interaction: collaboration is a fundamental component of knowledge sharing. Individuals or groups come together to share their expertise, perspectives, and ideas. Interaction facilitates a dynamic exchange of knowledge, allowing participants to learn from each other.
- Tacit and Explicit Knowledge: knowledge can be categorized into tacit and explicit forms. Tacit knowledge is often personal, based on individual experiences and insights. Explicit knowledge is codified and can be easily articulated. Knowledge sharing encompasses both tacit and explicit aspects.
- Learning and Development: knowledge sharing is closely linked to learning and development. Through the sharing of knowledge, individuals, organizations, and communities can enhance their understanding, skills, and problem-solving capabilities. It contributes to continuous improvement and adaptation.
- Community and Organizational Growth: in a broader context, knowledge sharing contributes to the growth of societies, organizations, and communities. As knowledge is disseminated and applied, it can lead to innovation, improved decision making, and the development of collective intelligence.
- Digital Platforms and Technology: in the modern era, technology plays a crucial role in knowledge sharing. For example, digital platforms, collaborative tools, and online forums can provide efficient means for individuals to share knowledge globally. This enables knowledge sharing on a large scale and mass level.
- Cultural and Social Context: the effectiveness of knowledge sharing is influenced by cultural and social factors. Cultural and social factors play a crucial role in shaping the dynamics of knowledge sharing within communities. These factors significantly influence how knowledge is disseminated, received, and utilized.
- Problem-Solving and Innovation: knowledge sharing is instrumental in problem-solving and innovation. By leveraging the collective expertise of a group, novel solutions can be created, and challenges can be addressed more effectively.
- Documentation and Knowledge Management: documenting knowledge is a crucial aspect of knowledge sharing. It ensures that insights are preserved and can be accessed by others. Knowledge management systems are often employed to organize and make knowledge readily available.

- Continuous Improvement: knowledge sharing is a catalyst for continuous improvement. By learning from successes and failures, individuals, organizations, and communities can refine their approaches, leading to ongoing growth and development.

Knowledge sharing is a dynamic process that fuels learning, collaboration, and progress. It is a cornerstone of human interaction and is essential for individual and collective advancements in various domains. Knowledge sharing is the lifeblood of learning communities. It transforms these spaces from passive environments into vibrant hubs where ideas flow freely, and learners actively contribute to the collective pool of understanding (Asrar-ul-Haq and Anwar 2016). In a learning community, knowledge is not confined to textbooks or lectures; it is a living, breathing entity shaped by the diverse perspectives and experiences of its members (learners). The effects of knowledge sharing within learning communities extend far beyond the immediate learners. As learners carry their enriched understanding into broader society, a ripple effect occurs. This dissemination of knowledge contributes to a more informed and interconnected world (Hwang et al. 2015; Shen 2015).

In the tapestry of education, knowledge sharing forms the intricate patterns that bind learning communities together. It transforms these communities into vibrant ecosystems where the collective wisdom of learners creates a reservoir of understanding that transcends learners' capabilities (Yilmaz 2017). The journey of learning becomes not just a solitary endeavor, but a collaborative voyage where the shared knowledge becomes a beacon illuminating the path toward a more informed and enlightened future. As technology advances, the future of knowledge sharing in learning communities presents exciting possibilities (Hall and Goody 2007). Artificial intelligence, virtual reality, and other relevant emerging technologies are poised to enhance the collaborative learning experience, providing new avenues for interaction and exploration. As learning communities continue to evolve, the methods and technologies employed for knowledge sharing must adapt and expand. For example, by combining traditional collaborative methods with innovative technology-driven approaches, educators can create dynamic learning environments that empower learners. The key lies in recognizing the diversity of relevant methods and technologies available and tailoring them to the unique needs of each learning community, fostering a culture where knowledge is created, shared, and continuously enriched (Stachová et al. 2020; Zamiri and Baqutayan 2012b).

### 1.2. Methods for Knowledge Sharing

The utilization of various methods for supporting knowledge sharing in learning communities is a strategic and multifaceted approach that enhances the educational and learning experience in numerous ways. The diversity of methods reflects the recognition that learners have different preferences, learning styles, and needs, and a one-size-fits-all approach is no longer effective in the present dynamic educational landscape (Chau et al. 2003). The importance of employing different methods lies in the ability to create a rich and engaging learning environment. By incorporating face-to-face or virtual communication, standardized operating procedures, documentation and manuals, collaboration platforms, social media, mentorship programs, and more, learning communities can cater to the diverse learning modalities and preferences of members (Szulczyńska and Majewska 2014).

The adaptability of these methods is crucial for accommodating the changing needs of members and the evolving demands of the educational and learning landscape. For example, face-to-face interactions foster personal connections and real-time feedback, while virtual communication enables flexibility and global connectivity. Standardized operating procedures ensure consistency, and documentation and manuals provide comprehensive resources for learners. Furthermore, training, workshops, and job rotation bring practical experiences to the learning process, allowing members to apply theoretical knowledge to real-world scenarios. Knowledge repositories provide a centralized hub for storing and accessing knowledge, ensuring that valuable insights are not lost over time (Koskinen and Pihlanto 2008; Jafari Navimipour and Charband 2016).

The importance of using these diverse methods is underscored by their collective ability to cater to the holistic development of members. They go beyond the traditional boundaries of education and learning, embracing a lifelong learning perspective. Moreover, they prepare learners not just with academic knowledge, but with practical skills, adaptability, and a global mindset—attributes essential in the interconnected and fast-paced world of the present (Islam 2012).

In essence, the integration of different methods for supporting knowledge sharing in learning communities reflects the commitment to fostering a dynamic, inclusive, and effective educational ecosystem. It acknowledges that learning is a multifaceted journey that involves collaboration, practical experiences, mentorship, and the integration of emerging technologies. By embracing this diversity of methods, learning communities can create an environment where knowledge is not only shared, but also actively applied, resulting in well-rounded, adaptable, and empowered individuals (Oliveira et al. 2022; Newk-Fon Hey Tow et al. 2012).

### 1.3. Knowledge Management in Learning Communities

Knowledge management (KM) within learning communities is a strategic process that involves identifying, capturing, organizing, and leveraging both explicit and tacit knowledge to enhance learning outcomes and community development. In the context of learning communities, knowledge management aims to create an environment where learners collaboratively create, share, develop, and apply knowledge to achieve common goals. It goes beyond the mere accumulation of knowledge and emphasizes the systematic utilization of knowledge for continuous improvement and innovation (Sekkal et al. 2019).

In learning communities, KM is closely tied to the idea of a shared knowledge base. This involves the creation of repositories, databases, and platforms that store relevant knowledge, learning resources, and insights generated by community members. These shared resources serve as a foundation for collaborative learning, allowing learners within the community to access and build upon the collective knowledge. KM in learning communities also involves creating mechanisms for effective communication and collaboration, ensuring that knowledge flows freely among community members (Gonzalvez et al. 2014).

KM provides the framework and tools for organizing and storing knowledge, making it accessible to learners within the community. This, in turn, promotes knowledge sharing by creating an infrastructure that supports collaboration and communication. Conversely, effective knowledge sharing contributes to the success of knowledge management initiatives. When learners freely exchange their expertise and insights, it enriches the knowledge repositories managed by the community, fostering a culture of continuous learning (Dei and Walt 2020).

As learning environments continue to evolve, knowledge management in learning communities becomes increasingly vital. The integration of technology, such as learning management systems and collaborative tools, plays a crucial role in facilitating KM practices within these communities. Overall, effective knowledge management in learning communities enhances the collective intelligence of the group, contributes to the development of innovative solutions, and fosters a culture of lifelong learning (Nesheim et al. 2011).

It is noteworthy that KM within a community involves several principles that govern how knowledge is requested and shared among learners. For example, the pull principle is centered on the idea that learners actively seek knowledge when they require it. In a learning community, this can manifest as learners accessing resources, engaging in discussions, or participating in collaborative projects based on their specific needs and learning objectives. Pull-based knowledge distribution empowers learners to take charge of their learning journey, ensuring that the knowledge they seek is relevant and timely. Conversely, the push principle involves proactively providing knowledge to learners based on their profiles, preferences, and learning histories. In a learning community, this can be achieved through personalized recommendations or notifications that highlight relevant content. The push principle recognizes that learners might not always be aware of what

they need, and by delivering curated knowledge, the community can enhance serendipitous learning experiences. It is particularly valuable in keeping learners informed about updates, emerging trends, or opportunities for collaboration. Both pull and push principles are integral to creating a dynamic knowledge ecosystem within a learning community. The pull principle respects the autonomy and agency of individual learners, while the push principle complements this by fostering a serendipitous learning environment. Striking a balance between these principles ensures that community members can access knowledge when needed, discover new insights serendipitously, and collectively contribute to the overall knowledge richness of the community. This dynamic interplay facilitates a continuous and adaptive knowledge flow, contributing to the growth and vibrancy of the learning community (Stefano 2022).

### 1.4. Technologies for Knowledge Sharing

The use of technologies for supporting knowledge sharing within learning communities signifies more than just a means of delivery—it represents a paradigm shift towards a comprehensive, productive, and adaptive educational ecosystem. As we navigate the ever-evolving landscape of education, embracing technology becomes not just a choice but a necessity, ensuring that learning remains relevant, accessible, and impactful for all (Hall and Goody 2007).

Technologies play a pivotal role in efficient knowledge sharing, enabling the rapid dissemination of news, changes, and updates. The multimodal nature of digital content accommodates various forms of learning, making education more inclusive and accessible. Additionally, the resource efficiency afforded by technologies, such as digital platforms, aligns with sustainability goals, contributing to environmentally conscious educational practices (Babu and Gopalakrishnan 2008).

The accessibility and flexibility afforded by digital technologies empower learners, breaking down the barriers of time and space. Learners can access a wealth of resources at their convenience, fostering a culture of continuous learning beyond traditional classroom settings. The diverse array of learning resources available through technology enriches educational content, catering to varied learning styles and preferences (Tiwari 2022).

Collaborative learning environments facilitated by online platforms and interactive technologies promote engagement and foster a sense of community. Members can connect with peers globally, bringing diverse perspectives to the forefront and preparing them for a connected world. Real-time feedback and assessment mechanisms provided by digital tools contribute to an iterative learning process, enhancing the understanding and mastery of subjects (Ilgaz and Aşkar 2013).

The personalization of learning experiences through adaptive technologies ensures that education is tailored to members' needs. This approach not only accommodates different learning styles, but also addresses the unique strengths and challenges of each member. Furthermore, the integration of data-driven decision-making processes allows members to refine their teaching strategies, improving the overall quality of education and learning (Karhukorpi and Cirhinlioglu 2022).

Beyond academic pursuits, technologies contribute to community building and networking through social media and online forums. These platforms foster the collaboration, discussion, and the exchange of ideas, creating vibrant online learning communities. The integration of digital technologies also prepares members for the demands of the digital age, cultivating digital literacy skills essential for success in learning communities (Brouwer and Jansen 2019).

To maximize the impact of knowledge sharing, learning communities must address challenges related to access and inclusivity. Efforts should be made to ensure that digital literacy barriers are overcome and resources are accessible to all learners, irrespective of their background or circumstances. Removing the challenges of knowledge sharing requires a comprehensive and strategic approach, such as creating a knowledge sharing culture, providing training and resources, facilitating communication and collaboration, addressing

cultural and social barriers, creating knowledge repositories, encouraging networking and communities of practice, promoting trust and psychological safety, measuring and evaluating the relevant issues, continuously improving them, and considering legal and ethical aspects. By implementing these strategies, communities can create an environment where knowledge sharing is not only encouraged but becomes an integral part of the community culture (Muqadas et al. 2017; Matsuo and Aihara 2022).

Considering the above points, the motivation behind conducting this systematic literature review of the methods and technologies for supporting knowledge sharing within learning communities stems from the evolving landscape of education and the need to understand how better (related and effective) methods and technologies synergize to facilitate and enhance knowledge sharing. In an era marked by unprecedented global connectivity and progressive collaboration applications, the need to comprehensively explore and evaluate the diverse approaches employed within learning communities becomes crucial. This review attempts to bridge the existing gaps in knowledge by synthesizing the relevant literature and providing insights into the strengths and challenges associated with different methods and technologies. By providing an overview of the current state of the art, the review seeks to empower educators, researchers, and decision makers with evidence-based information. On the other hand, despite the researchers being increasingly intrigued by the phenomenon of knowledge sharing, there is a limited body of literature that comprehensively outlines and presents effective methods and technologies for sharing knowledge within learning communities. Thus, the main objectives of this work are to (a) conduct a systematic literature review of the related studies and (b) identify and document some potential methods and technologies that support knowledge sharing in learning communities, aiming to provide a robust, evidence-based synthesis of the existing knowledge, guide future research, and contribute to evidence-informed decision making in the contexts of education and community learning.

The remainder of this study is organized as follows: in Section 2, the research method used for this study is briefly explained. In Section 3, the identified methods and technologies for supporting knowledge sharing in learning communities are presented. In Section 4, a discussion is developed about the findings of this survey. The paper ends with some concluding remarks and a brief analysis of possible future work.

## 2. Systematic Literature Review

A systematic literature review is a methodical and comprehensive approach to reviewing the existing scholarly literature on a specific topic or research question. It involves a structured and systematic process to identify, evaluate, and synthesize relevant studies and publications from academic databases and other sources. Systematic literature reviews are commonly used in academic research to provide a comprehensive and unbiased overview of the existing evidence for a particular topic. They are valuable for informing decision making, identifying gaps in the literature, and guiding future research directions.

From this perspective, for this study, a systematic literature review was adopted to identify, analyze, integrate, and interpret the findings of multiple works. To properly guide the study, the following two research questions were formulated:

Q1. How can knowledge sharing be supported within learning communities?

Q2. Which methods and technologies are best used for the type of learning community or size of the community?

To answer these research questions, the following hypotheses were adopted:

**Hypothesis 1 (H1).** *Knowledge sharing within learning communities can be supported if potential and effective methods and technologies are applied.*

**Hypothesis 2 (H2).** *The selection of methods and technologies for knowledge sharing within learning communities depends on factors, such as the unique needs and attributes of each learning community, learner, and knowledge to be shared.*

We followed the procedure described by the Preferred Reporting Items for Systematic Review and Meta-Analyses (PRISMA). PRISMA refers to a set of guidelines and a checklist designed to enhance the transparency and completeness of reporting in systematic reviews and meta-analyses. PRISMA provides a standardized framework for authors to report their review processes and findings. It was developed to improve the quality and clarity of these research publications (Moher et al. 2009).

Following PRISMA, this section explains how the methods and technologies for knowledge sharing in learning communities are identified, selected, and critically appraised in this study. The ensuing subsections delineate the search process, encompassing the paper selection process. Then, the approach for reviewing and analyzing selected articles is presented.

### 2.1. Article Selection Process

The strategy for selecting papers encompassed three main stages:

- Stage 1 (identification): in this stage, the automated search focused on several keywords, namely, 'methods and technologies for knowledge sharing in learning communities', 'methods and technologies for knowledge sharing', 'supporting knowledge sharing in learning communities', 'facilitators of knowledge sharing', 'knowledge management in learning communities', and 'knowledge management practices in learning communities'. To search and choose the relevant (English language) studies for the survey, the main engineering and computer science databases, such as ScienceDirect, SpringerLink, Web of Science, and IEEE, were selected. Exclusions were made for working papers, reports, errata, editorial notes, and commentaries, resulting in the identification of a total of 217 studies.
- Stage 2 (review): this stage focused on the structured process of screening and evaluating the titles, keywords, abstracts, and conclusions of the identified studies for relevance. Consequently, a total of 123 studies were selected for full reading.
- Stage 3 (includes): in this stage, a comprehensive examination was conducted on the 54 selected studies (8 books, 33 journals, and 13 conferences) that met the inclusion criteria established for the review. This process involved analyzing the content of each selected study in detail to extract the relevant information, identify patterns, synthesize the findings, and draw conclusions. This stage is presented in detail in Section 2.2.

Figure 1 demonstrates the flowchart corresponding to this review.

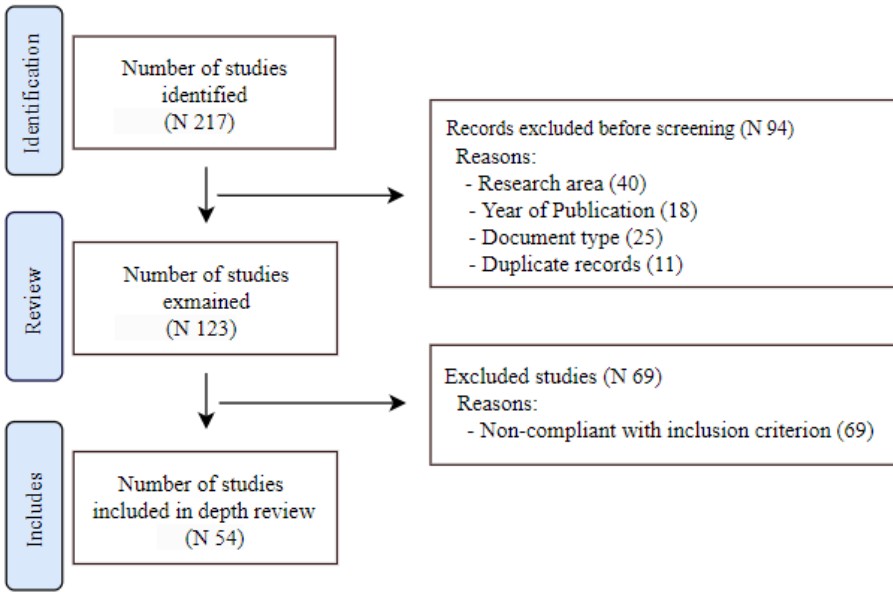

**Figure 1.** PRISMA flowchart of the inclusion and exclusion criteria.

Figure 2 depicts the distribution of the studies (selected in stage 3) by year of publication. It shows the studies published in 2010–2023.

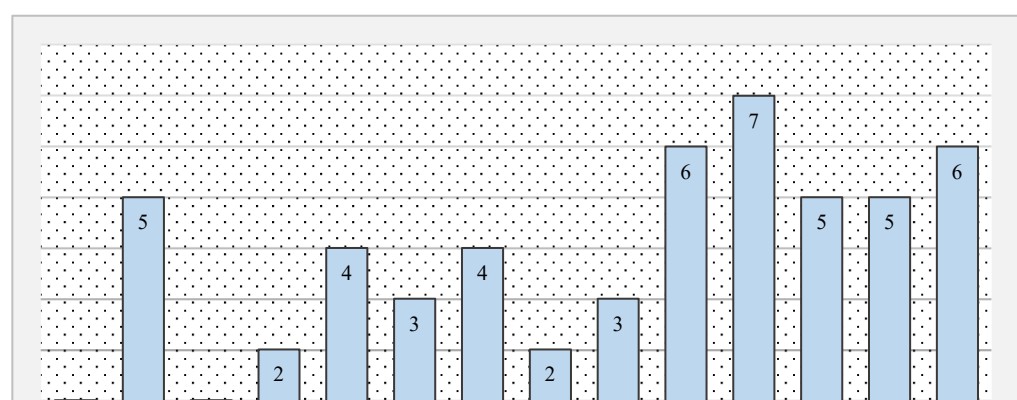

**Figure 2.** Distribution of studies by year of publication.

### 2.2. Reviewing and Analyzing the Selected Articles

The review and analysis phase in the systematic literature review involved a structured examination of the selected studies to draw meaningful insights and conclusions. The following points outline the steps we took to review and analyze the selected studies in stage 3 (includes):

- Data Extraction: having considered the familiarization (reading the titles, keywords, abstracts, and conclusions to understand the scope and focus of each study) and also taken into account the inclusion and exclusion criteria in the process of the study selection (presented in Section 2.1), we proceeded with systematically extracting the relevant information from each study. This included details, such as the study goals, design, methodology, sample size, key findings, and any other pertinent information.
- Quality Assessment: in this step, we assessed the quality of each study based on the study design, methodology, and potential biases. This step was crucial for gauging the reliability and validity of the evidence.
- Thematic Analysis: in this step, we identified and categorized the key themes or topics emerging from studies. We grouped similar findings to reveal patterns, trends, or commonalities across the literature.
- Synthesis of Results: in this step, we summarized the findings and results of each study. We provided a synthesis of the main findings and results, comparing and contrasting the outcomes of different studies. The output of this step (which is the main contribution of this work) is presented in Section 3.
- Identify Gaps and Limitations: in this step, we identified the gaps in the existing literature. We also revealed the limitations of individual studies and the overall body of evidence (which are presented in Section 4.1).
- Recommendations for Future Research: in this step, we suggested the areas for future research (which are presented in Section 4.2) based on the gaps identified during the review. This contributes to the ongoing development of the field.

By following these steps, we (a) validated the selection of studies and (b) synthesized the collective knowledge, identified gaps, and contributed to the overall body of academic literature. These were crucial steps for ensuring the rigor, reliability, and practical relevance of the research findings.

### 3. Identified Methods and Technologies for Supporting Knowledge Sharing in Learning Communities

In this section, we present the identified methods and technologies for supporting knowledge sharing in learning communities, offering concise explanations for each. These methods and technologies not only support knowledge sharing, but also play a crucial role in fostering effective learning environments and promoting individual and collective growth. In other words, they are instrumental in creating vibrant, collaborative, and adaptive educational environments that empower learners and contribute to the broader goals of education and professional development.

*3.1. Identified Methods for Supporting Knowledge Sharing in Learning Communities*

Learning communities, with their diverse members and collective pursuit of knowledge, represent dynamic ecosystems where knowledge flows in myriad ways. The significance of employing different methods for supporting knowledge sharing within these communities cannot be overstated, that is, it lies in the richness it brings to the learning experience. By acknowledging the diversity of learning styles, preferences, and constraints, communities can create inclusive environments that cater to the needs of all members. The synergy between traditional and new methods fosters engagement, collaboration, and innovation. As we navigate the ever-evolving landscape of education, embracing a diversity of methods not only becomes a strategy but a philosophy that recognizes the multifaceted nature of learning communities and the unique contributions each participant brings to the collective pursuit of knowledge. Knowledge sharing in learning communities can take various forms, and the choice of methods often depends on the context, audience, and nature of the knowledge being shared (Rakov and De Ridder 2022; Jafari Navimipour and Charband 2016; Balle et al. 2016). In the following points, the identified methods that can be used for knowledge sharing in learning communities are presented:

- Providing Face-to-Face or Virtual Communication: both face-to-face and virtual communication have their advantages and are applicable to different contexts. The choice often depends on factors, such as the nature of the interaction, the need for immediacy, and the geographical dispersion of community members. In the interconnected world of the present day, a blend of both modes is common, allowing for flexibility and effective communication across various situations. Face-to-face communication can be utilized in scenarios such as team-building workshops, practical skills development, sensitive discussions, networking events, and immediate feedback. Virtual communication can be utilized in scenarios, such as global communities, online courses and webinars, asynchronous collaborations, document sharing and collaboration, regular updates and announcements, and recordings and archives (Aljuwaiber 2019).

- Providing Documentation and Manuals: these factors play a crucial role in supporting knowledge sharing in learning communities. These methods involve creating organized, accessible, and comprehensive resources that serve as references for learners. By employing documentation and manual methods, learning communities can create a structured and accessible knowledge sharing environment, empowering learners to navigate, contribute to, and benefit from the collective wisdom of the community. Documentation can be utilized in scenarios such as orientation programs, certification courses, documentation of case studies, learning communities with specific compliance, and troubleshooting technical issues. Manuals can be utilized in scenarios such as welcoming new members, skill development, documenting best practices, outlining community guidelines and codes of conduct, and navigating and using the community platform (Gehrke and Hasan 2020).

- Providing Standard Operating Procedures (SOPs): SOPs are detailed, written instructions that provide a step-by-step guide on how to perform a specific task or process within a community. SOPs are designed to ensure consistency, quality, and compliance with regulatory standards. They serve as a reference document for learners, outlining the prescribed methods, responsibilities, and protocols for various operations.

By adopting these procedures, learning communities can create a robust system of standard operating procedures that not only facilitate knowledge sharing, but also contribute to the overall efficiency and effectiveness of community operations. SOPs can be utilized in scenarios such as community structure, membership policies, content quality, evaluation processes, internal and external communications, workshops and webinars, annual conferences, and knowledge repositories (Pangil and Nasurddin 2013).

- Providing Mentorship Programs: these programs are structured relationships where an experienced individual (mentor) provides guidance, advice, and support to a less experienced individual (mentee). These programs are powerful mechanisms for knowledge sharing within communities. Implementing these mentorship program methods creates a dynamic and supportive environment within learning communities, fostering the exchange of knowledge, skills, and experiences among community members. Mentorship programs can be utilized in scenarios such as developing specialized skills, career advancement, moving from academia to industry or changing career paths, joining new members, collaborative projects, research or academic initiatives, and leadership development within learning communities (Al Bukhari Marzuki 2021).

- Providing Communities of Practice (CoPs): this is a group of members who share a common interest, profession, or expertise, and come together to interact regularly, learn from each other, and collaboratively develop knowledge within that shared domain. CoPs are recognized for their role in facilitating knowledge sharing, creating a sense of belonging, and promoting continuous learning. By implementing these methods, communities of practice can become vibrant learning ecosystems that actively support knowledge sharing and collaborative learning among their members. CoPs can be utilized in scenarios such as professional development, industry-specific communities, interdisciplinary learning, project-based collaborations, fostering innovation within learning communities, and problem-solving communities (Matsuo and Aihara 2022).

- Providing Training Sessions, Seminars, Webinars, Conferences, and Workshops: these are integral components of community learning and development methods. They provide a structured and interactive environment for knowledge sharing, skill development, and continuous improvement. Training sessions can be utilized, for example, for industry-specific training. Seminars can be utilized for continuing education seminars. Webinars can be utilized for remote learning communities. Academic conferences can be utilized for research communities. Workshops can be utilized for skill enhancement workshops and technical skill workshops (Abinash and Subaveerapandiyan 2021).

- Job Rotation: this is a method for supporting knowledge sharing in learning communities by providing members with opportunities to experience different roles within a community. This practice helps the transfer of knowledge, skill development, and the creation of a more versatile and adaptable member. When job rotation is strategically implemented, it can be a powerful tool for knowledge sharing, talent development, and organizational agility. Job rotation can be utilized in scenarios such as leadership development programs, corporate training programs, cross-functional teams, skills enhancements in multidisciplinary communities, and community-based learning initiatives (Al-khalidi and Jassim 2018).

- Storytelling: this involves using narrative techniques to convey information, insights, or experiences compellingly and memorably. This method harnesses the power of storytelling to make complex concepts more relatable and easier to understand. By incorporating storytelling methods, learning communities can create a rich tapestry of experiences, lessons, and insights that enhance knowledge sharing and foster a sense of community among its members. Storytelling can be utilized in scenarios such as cultural and diversity education, corporate training and onboarding, team building and collaboration, community-based learning initiatives, and conflict resolution and mediation workshops (Park et al. 2020).

- Gamification: this involves integrating elements of game design and mechanics into non-game contexts, such as learning and knowledge sharing, to enhance engagement,

motivation, and participation. The idea is to leverage the inherent appeal of games to make activities more enjoyable and encourage learners to actively participate in the acquisition and sharing of knowledge. Gamification can be utilized in scenarios such as employee training and onboarding, language learning programs, STEM education, professional development and skill building, diversity and inclusion training, and soft skills development (Friedrich et al. 2020).

- Feedback Seeking: this is an essential method for gathering input, insights, and opinions from members of a community. When applied to knowledge sharing, this method becomes valuable for assessing the effectiveness of knowledge sharing initiatives, understanding the needs of learners, and identifying areas for improvement. By regularly incorporating feedback seeking, learning communities can create a culture of continuous improvement, ensuring that knowledge sharing initiatives align with the needs and preferences of their members. Feedback seeking can be utilized in scenarios such as professional development needs assessments, the effectiveness of training programs, community engagement and satisfaction, community planning and decision making, and post-project evaluations in project-based learning (Cheng 2017).

- Fostering Effective Collaborations: this involves encouraging collaborations across different learning communities, different disciplines, expertise areas, or domains. This approach facilitates the exchange of diverse perspectives and knowledge. By implementing cross-collaboration methods, learning communities can break down silos, promote knowledge exchange between disciplines, and create a rich environment for interdisciplinary learning and innovation. Fostering effective collaborations can be utilized in scenarios such as project-based learning initiatives, community-based initiatives, global learning communities, and professional development and skill-building programs (Ahmed et al. 2016).

- Providing Learning Circles: these form small learning circles or study groups focused on specific themes or subjects. Learners collaborate in a structured environment to expand their understanding of particular topics. Learning circles can be utilized in scenarios such as professional development, online course communities, community-based learning initiatives, stem education programs, and teacher professional learning communities (Nazeer and Isani 2021).

- Peer Review: this implements a peer review system where community members can share their work for constructive feedback. This promotes collaborations and helps refine ideas and knowledge. Peer reviews can be utilized in scenarios such as language learning communities, online course communities, professional development programs, project-based learning, teacher professional learning communities, and online writing communities (Bergquist et al. 2001).

- Providing Discussion Boards/Panels: this method organizes discussion boards/panels with experts or community leaders. These sessions allow for the in-depth exploration of specific topics and encourage participation from the learners. Discussion boards/panels can be utilized in scenarios such as academic conferences, professional development workshops, business and entrepreneurship communities, technology and innovation hubs, and environmental and sustainability communities (Lee et al. 2006; Zamiri et al. 2023).

- Providing E-newsletters: this method distributes regular newsletters containing updates, member highlights, and valuable content. E-newsletters are a convenient way to share knowledge and information directly with community members. E-newsletters can be utilized in scenarios such as professional development communities, research and academic groups, community engagements and discussions, language learning communities, and community recognition and spotlights (Wang and Chien 2019).

- Providing FAQs: this method has a frequently asked questions section, addressing common queries within the community. This can serve as a quick reference for learners. FAQs can be utilized in scenarios such as onboarding and orientation,

technical support, updates and changes, resource access, and community features (Musbah et al. 2019).

- Providing Collaborative Problem-Solving Sessions: this method hosts sessions dedicated to collaborative problem-solving. Learners can highlight the challenges they are facing, and the community collectively works on finding solutions. Collaborative problem-solving sessions can be utilized in scenarios such as interdisciplinary projects, group learning projects, community challenges, community improvement initiatives, and critical discussions (Stevens et al. 2005).

- Providing User-Generated Content: this method encourages learners to contribute user-generated content, such as articles, case studies, or tutorials. This promotes a culture of shared expertise and knowledge. User-generated content can be utilized in scenarios such as project-based learning communities, discussion forums and Q&A platforms, language learning communities, innovation and creativity hubs, and community challenges and competitions (Seneviratne and Hewakuruppuge 2023).

- Knowledge Cafes: this method is an informal gathering where learners engage in conversations around specific topics. This approach encourages free-flowing discussions. Knowledge cafes can be utilized in scenarios such as professional development communities, cross-disciplinary learning networks, problem-solving communities, and learning and teaching communities (Hamidian 2015).

- Providing Crowdsourced Projects: this method initiates crowdsourced projects where community members collaborate on creating valuable resources, tools, or content. Crowdsourced projects can be utilized in scenarios such as problem-solving initiatives, content creation and curation, community engagement and decision making, community building and engagement, and skills development and training (Martinez 2015).

The effectiveness of knowledge sharing methods often depends on the community culture, the preferences of the community, and the nature of the knowledge being shared. Using one or some of the above-mentioned techniques in learning communities is a strategic approach to creating an inclusive, engaging, and effective learning environment that caters to the diverse needs and preferences of its members. These methods can support the overarching goal of facilitating collaborative learning experiences and meaningful knowledge sharing.

### 3.2. Identified Technologies for Supporting Knowledge Sharing in Learning Communities

The integration of technology into learning communities is a transformative force that goes beyond mere knowledge dissemination. It catalyzes collaborations, enhances accessibility, promotes engagement, and personalizes learning experiences. The interconnectedness facilitated by technology creates a dynamic and inclusive environment where knowledge is not confined to textbooks but is a collaborative and evolving entity shaped by the contributions of diverse learners. As we embrace the digital era, the synergy between technology and knowledge sharing is poised to redefine the landscape of education, making learning more accessible, engaging, and meaningful for members of learning communities (Pang et al. 2020; Zafar Yaqub and Alsabban 2023; Naeem 2019; Arsenijevi et al. 2011; Huang et al. 2023). In the following section, we provide a list of identified technologies that can be used for supporting knowledge sharing within learning communities. For each technology, some popular and commonly used examples are listed, coupled with their main related characteristics. The examples were identified through the literature review and the parallel search of related websites and platforms dedicated to educational technology, knowledge management, and learning communities.

- Learning Management Systems (LMSs): these are software applications or platforms designed to facilitate the administration, documentation, tracking, reporting, and delivery of educational courses, training programs, or learning and development programs. They also play a crucial role in knowledge sharing within learning communities by providing a centralized and organized platform for managing educational content and fostering collaborations. LMSs can be utilized in scenarios such as corporate training,

online courses and e-learning, compliance training, blended learning environments, government training initiatives, and nonprofit learning communities (Cuéllar et al. 2011). Here are some of the best LMSs that can be used for knowledge sharing:

- Moodle:
    - ○ It is an open source learning platform.
    - ○ It is highly customizable with a large, active community.
- Canvas by Instructure:
    - ○ It is known for its user-friendly interface.
    - ○ It offers various features, like discussion boards, quizzes, and collaborative tools.
- Blackboard Learn:
    - ○ A widely used LMS in higher education.
    - ○ Provides tools for communication, content delivery, and assessment.
- D2L Brightspace:
    - ○ It focuses on improving the learning experience.
    - ○ It is used in K-12, higher education, and corporate training.
- Schoology:
    - ○ It is popular in K-12 education.
    - ○ It emphasizes collaboration and engagement.
- Google Classroom:
    - ○ It is part of the Google Workspace for Education.
    - ○ It integrates seamlessly with Google Drive and other Google tools.
- Totara Learn:
    - ○ It is an open source LMS based on Moodle.
    - ○ It offers extended features for corporate learning and development.
- Adobe Captivate Prime:
    - ○ It is part of Adobe's eLearning suite.
    - ○ Suitable for corporate training.
- Collaboration Platforms: digital tools and systems designed to facilitate communication, teamwork, and knowledge sharing within a community. These platforms are designed to enhance teamwork, streamline knowledge sharing, and improve overall productivity. Collaboration platforms can be utilized in scenarios such as team projects and assignments, innovation and ideation, community learning initiatives, community engagement, and volunteer and community services (Aggarwal et al. 2011). Here are some examples of collaboration platforms commonly used for knowledge sharing:
- Microsoft Teams:
    - ○ It combines chat, video conferencing, file sharing, and application integration.
    - ○ It is suitable for both small and large communities.
- Slack:
    - ○ It enables real-time communication through channels and direct messages.
    - ○ It integrates with various third-party apps and services.
- Asana:
    - ○ It supports task management, team communication, and project tracking.
    - ○ It can be used for organizing and coordinating work within communities.
- Trello:
    - ○ It organizes work using boards, lists, and cards.
    - ○ It facilitates collaboration and task tracking.

- Jira:
  - It manages and tracks issues and projects.
  - It offers advanced customizations for agile project management.
- Confluence:
  - It is a wiki-style collaboration platform.
  - It allows learners to create, share, and collaborate on content.
- Basecamp:
  - It is a project management and collaboration software.
  - It provides tools for task management, document sharing, and communication.
- Yammer:
  - It facilitates communication and collaboration across a community.
  - It supports file sharing and group discussions.
- Wikis: collaborative websites or platforms that enable learners to create, edit, and link web pages easily. They serve as powerful tools for knowledge sharing in community learning environments, providing a space for community members to contribute, modify, and organize knowledge collectively. Wikis can be utilized in scenarios such as collaborative documentation, course content creation, community portals, community-driven resources, and knowledge repositories (Arazy et al. 2016). Here are some examples of Wikis commonly used for knowledge sharing:
  - MediaWiki:
    - It is one of the most well-known wiki platforms, known for its flexibility and scalability.
    - Many communities use it for internal documentation.
  - Confluence:
    - It is a collaborative wiki tool developed by Atlassian.
    - It is commonly used by businesses for project collaborations, document sharing, and knowledge management.
  - DokuWiki:
    - It is a simple-to-use and highly versatile wiki platform.
    - It is often used for personal or small-team projects and is known for its ease of installation and maintenance.
  - Tiki Wiki CMS Groupware:
    - It is a comprehensive open source wiki-based content management system.
    - It is used for a wide range of applications, including collaboration, project management, and education.
  - TWiki:
    - It is a structured wiki application that facilitates team collaboration.
    - It is used for project management, documentation, and knowledge sharing in various communities.
  - XWiki:
    - It is an open source wiki platform with advanced features for collaboration.
    - It is used for creating collaborative solutions, intranet applications, and knowledge management systems.
  - PBworks:
    - It is a hosted collaboration solution that includes wiki functionality.

- ○ It is used for business collaboration, project management, and creating shared workspaces.
    - Wikidot:
        - ○ It is a wiki-hosting service that allows learners to create and edit wikis.
        - ○ It is used for a variety of purposes, including personal wikis, project collaboration, and educational resources.
- Video Conferencing Tools: video conferencing tools have become essential in community learning for facilitating communication, collaboration, and knowledge sharing among members, especially in remote or online learning settings. Some examples of video conferencing tools (e.g., Microsoft Teams, Zoom, and Cisco Webex Teams) have the features of collaboration platforms and are presented in this group of technologies. Video conferencing tools can be utilized in scenarios such as remote learning, collaborative group projects, tutoring and mentoring, language exchange programs, and online study groups (Nilsen 2011). Here are the other popular video conferencing tools that can be used for knowledge sharing:
    - Google Meet:
        - ○ It allows video meetings with up to two learners.
        - ○ It integrates with Google Calendar for easy scheduling.
    - Skype for Business:
        - ○ It supports video and audio conferencing, instant messaging, and screen sharing.
        - ○ It is suitable for smaller- to medium-sized meetings.
    - GoToMeeting:
        - ○ It can be used for screen sharing, meeting recording, and integrations with productivity tools.
        - ○ It is suitable for communities of various sizes.
    - BlueJeans by Verizon:
        - ○ It supports integrations with productivity and collaboration tools.
        - ○ It is suitable for large-scale virtual events.
    - Adobe Connect:
        - ○ It offers interactive features, like polls, chat, and breakout rooms.
        - ○ It is suitable for training sessions and educational purposes.
    - Jitsi Meet:
        - ○ It allows learners to set up virtual meetings with ease.
        - ○ It supports features, like screen sharing and integration with other tools.
    - RingCentral:
        - ○ It is a cloud-based communication platform.
        - ○ It offers video conferencing, team messaging, and file sharing.
    - Zoho Meeting:
        - ○ It enables online meetings and webinars.
        - ○ It integrates with other Zoho productivity tools.
- Social Media and Networking Platforms: these serve as dynamic spaces for knowledge exchange, idea generation, and collaboration. They offer diverse formats for content sharing, ranging from text and images to videos and audio, making them versatile tools for knowledge sharing across different contexts. The choice of social media platforms depends on the preferences of the community, the nature of the content being shared, and the desired level of interaction and collaboration. It is common for learning communities to leverage a combination of these platforms based on their specific needs and goals. Social media and networking platforms can be utilized in scenarios such as community building, peer support, global collaboration, instructor

engagement, and live sessions and webinars (Naeem 2019). Here are some examples of social media and networking platforms commonly used for knowledge sharing:

- LinkedIn:
    - ○ It is a professional networking platform where learners can connect with peers, join professional groups, and share related insights.
    - ○ Professionals, organizations, and communities use LinkedIn for networking, sharing news, and participating in discussions.
- Twitter:
    - ○ It is a microblogging platform that allows learners to share short messages (tweets) with a wide audience.
    - ○ It is used for real-time updates, sharing links to articles and resources, and participating in conversations using hashtags.
- Facebook:
    - ○ It is a social networking platform where learners can create profiles, connect with friends and peers, and join groups.
    - ○ Groups and pages on Facebook can be used for knowledge sharing within specific communities or industries.
- Instagram:
    - ○ It is a visual-centric platform for sharing photos and short videos.
    - ○ It is often used by communities to share visual content, infographics, and short educational videos.
- YouTube:
    - ○ It is a video-sharing platform where learners can upload, share, and view videos.
    - ○ Communities and learners can create and share educational videos, tutorials, and webinars.
- Reddit:
    - ○ It is a platform that consists of communities (subreddits) where learners can share links, and images, and also engage in discussions.
    - ○ It is used for discussions on specific topics, sharing articles, and asking questions.
- Quora:
    - ○ It is a question-and-answer platform where learners can ask questions and receive answers from experts or individuals with relevant knowledge.
    - ○ It is used for seeking and sharing knowledge on various topics.
- Pinterest:
    - ○ It is a visual discovery platform where learners can find and save ideas for various projects and interests.
    - ○ It is used for curating and sharing visual content related to specific topics.
- Blogs and Microblogging: blogs are online platforms where learners or communities regularly post written content, often in a journal-like format. Blog posts can cover a wide range of topics, including industry trends, tutorials, opinions, and more. Microblogging involves sharing short, concise messages with a focus on brevity. Platforms for microblogging often limit the length of learners' posts to a small number of characters. Blogs and microblogging platforms are powerful tools for knowledge sharing in community learning environments. They provide a space for learners to share insights, reflections, and resources, fostering a sense of community and collaborative learning. Blogs and microblogging can be utilized in scenarios such as collaborative writing projects, discussion and debate, peer feedback and reviews,

collaborative problem-solving, and global connectivity (Hoong and Lim 2014). Here are some examples of blogs and microblogging commonly used for knowledge sharing:

- Blogs:
- WordPress:
  - ○ It allows learners to create and manage websites and blogs.
  - ○ It supports various media types, including text, images, and videos.
- Medium:
  - ○ It is designed for writers to share their stories and insights. Medium also has a built-in audience, making it easier for content to reach a broader readership.
  - ○ It allows learners to clap for and respond to articles.
- Blogger:
  - ○ Learners can create blogs with a Blogger subdomain or connect a custom domain.
  - ○ It offers various templates for blog customization.
- Microblogging:
- Tumblr:
  - ○ It allows learners to post multimedia content in a short-form blog format.
  - ○ It combines elements of blogging with social networking.
- Twitter:
  - ○ It facilitates real-time communication and knowledge sharing.
  - ○ Learners can amplify content by retweeting (sharing) tweets to their followers.
- Discussion Forums: also known as online forums or message boards. They provide a structured space for learners with similar interests or expertise to discuss various topics, ask questions, share insights, and collaborate on projects. Choosing the right platform depends on factors such as the community's needs, preferences, and the desired level of customization. Discussion forums can be utilized in scenarios such as collaborative problem-solving, peer-to-peer support, instructor–student interactions, community building, and project collaboration (Boh 2014). Here are some examples of discussion forums that can be used for knowledge sharing:
  - Reddit:
    - ○ It is a platform where learners can submit content, such as text posts, links, and images, and participate in discussions within various communities known as subreddits.
    - ○ It relies heavily on user-generated content.
  - Stack Exchange:
    - ○ It hosts communities focused on specific topics, where learners can ask questions, provide answers, and engage in discussions.
    - ○ It accumulates a vast repository of knowledge.
  - Quora:
    - ○ It is designed for learners to ask questions and receive answers from individuals with knowledge in various fields.
    - ○ It covers a wide range of topics, allowing learners to find knowledge and engage in discussions on virtually any subject.
  - phpBB:
    - ○ It provides a platform for discussions on a wide range of topics.
    - ○ Learners can format their posts using BBCode or HTML, allowing for rich text formatting and media embedding.

- Podcasts: these are a form of digital media that involves the creation and distribution of audio or video content in an episodic series. They have become a popular medium for knowledge sharing in community learning environments due to their accessibility, convenience, and ability to convey knowledge engagingly. Podcasts can be utilized in scenarios such as supplemental learning material, expert interviews, language learning, student-created content, discussion and debate, and community building (Louadi et al. 2023). Here are some notable podcasts that contribute to knowledge sharing:
  - TED Talks Daily:
    - ○ Daily episodes featuring TED Talks on a wide range of topics, including science, technology, education, and personal development.
    - ○ TED often encourages interactions and discussions around its talks.
  - The Tim Ferriss Show:
    - ○ Tim Ferriss interviews world-class performers from different fields, exploring their strategies, routines, and lessons learned.
    - ○ It includes discussions about books and book recommendations.
  - The Knowledge Project with Shane Parrish:
    - ○ Shane Parrish explores the art of learning and decision making through conversations with experts in various domains.
    - ○ "The Knowledge Project" centers around the concept of continuous learning.
  - The EdSurge Podcast:
    - ○ Discussions on the intersection of technology, education, and the future of learning, exploring the innovations and challenges in education.
    - ○ It promotes learners' collaboration, creativity, and critical thinking.
- Virtual Reality (VR): refers to a computer-generated environment that simulates a realistic experience. It immerses users in a completely virtual world, typically using headsets or goggles. In learning communities, VR can be used to create simulated environments for training and education. For example, medical students can virtually explore the human body or historical events can be recreated for a more immersive history lesson. VR can be utilized in scenarios such as language learning, soft skills training, simulations for emergency response training, job training simulations, and collaborative projects and community building (Gurian et al. 2023). Here are some examples of how VR is applied for knowledge sharing:
  - Virtual Tours and Field Trips:
    - ○ It allows learners to take virtual tours of historical sites, museums, or landmarks.
    - ○ It provides learners with the opportunity to explore places they might not otherwise visit.
  - Medical Training Simulations:
    - ○ In the field of medicine, it is used for training simulations.
    - ○ Medical students can practice surgeries, diagnose conditions, and interact with virtual patients in a realistic environment.
  - Language Learning:
    - ○ Language learners can use it to simulate real-world language scenarios.
    - ○ It allows learners to practice conversations, navigate foreign locations, and enhance language skills in a contextual setting.
  - Collaborative Learning Spaces:
    - ○ It offers collaborative learning spaces where learners from different locations can come together in a shared virtual environment.
    - ○ It is beneficial for remote or online education.

- Augmented Reality (AR): overlays digital knowledge and information onto the real-world environment. Unlike VR, AR does not create a fully immersive alternate reality; instead, it enhances the learner's perception of the real world. AR can be applied in various educational scenarios. For instance, pointing a device at a textbook can trigger additional knowledge, information, or animations related to the content. In professional settings, AR can be used for on-the-job training, providing real-time guidance and information. AR can be utilized in scenarios such as language learning, stem education, museum and exhibition learning, collaborative learning projects, outdoor education, and gamified learning (Del Amo et al. 2018). Here are examples of how AR is used for knowledge sharing:
    - Education:
        - It is used in educational apps to provide interactive learning experiences.
        - Learners can use AR to explore 3D models, historical timelines, and scientific concepts.
    - Training Simulations:
        - It is used for creating realistic training simulations in various communities.
        - It allows learners to practice procedures in a controlled environment.
    - Remote Assistance:
        - It is used for remote assistance.
        - Experts can provide guidance and instructions by overlaying knowledge and information onto the learners' field of view.
    - Collaborative Workspaces:
        - It is used for collaborative work environments.
        - Learners in different locations can interact with shared digital content as if it exists in the same physical space.
- E-learning Platforms: these platforms are digital tools that facilitate the delivery of educational content and training programs through electronic means. These platforms often include features such as online courses, interactive content, assessments, and collaboration tools. E-learning platforms can be utilized in scenarios such as massive open online courses (MOOCs), blended learning environments, adaptive learning, global collaboration, and assessment and feedback (Wang and Wen 2011). Here are some examples of popular E-learning platforms for knowledge sharing:
    - Moodle:
        - It includes collaboration tools, such as forums, wikis, and discussion boards, fostering interaction and knowledge sharing among learners and instructors.
        - It supports multimedia elements, allowing the integration of videos, audio files, and other interactive content to enhance the learning experience.
    - edX:
        - It offers a diverse array of courses, covering various subjects and disciplines, including science, technology, engineering, mathematics, humanities, and more.
        - Courses on edX often include interactive elements, such as quizzes, discussions, and collaborative projects.
    - Coursera:
        - Many courses on Coursera have instructor-led sessions, providing a structured learning experience with regular deadlines and opportunities for interaction with instructors and peers.
        - It utilizes learning analytics to track learner progress and engagement, providing insights to both learners and instructors.
    - Skillshare:

- It is a platform that allows creators to offer classes in various creative disciplines, such as design, photography, writing, and more.
- Courses on Skillshare often consist of short-form video content, making it easy for learners to consume and apply knowledge in bite-sized segments.

- Khan Academy:
  - It offers free online courses, lessons, and practice in various subjects, primarily targeting K-12 students.
  - The platform fosters a sense of community through discussion forums where learners can ask questions, share insights, and engage with each other.

- Udemy:
  - Udemy is an online learning platform that allows experts in various fields to create and sell courses.
  - Learners can explore new topics and acquire skills throughout their lives, contributing to a culture of ongoing education.

- Document Management Systems (DMSs): play a crucial role in knowledge sharing within community learning environments by providing a centralized platform for creating, storing, organizing, and accessing educational content. These systems enhance collaborations, streamline workflows, and ensure efficient document management. DMSs can be utilized in scenarios such as collaborative project documentation, course materials management, faculty collaboration and communication, collaborative learning projects, and remote learning documentation (Baban and Mokhtar 2010). Here are some examples of popular document management systems for knowledge sharing:

  - Microsoft SharePoint:
    - Learners can create, share, and collaborate on documents in real time.
    - Communities can configure SharePoint to allow external sharing, facilitating collaboration with external partners, clients, or vendors while maintaining control over permissions.

  - Google Drive:
    - It is a cloud-based file storage and synchronization service.
    - Learners can share files and folders with others, and permissions can be set to control who can view, edit, or comment on documents.

  - Dropbox Business:
    - Dropbox Business is a cloud-based solution that enables teams to store and share files securely.
    - Learners can share files and folders with colleagues both within and outside the community.

  - M-Files:
    - It is a document management solution that offers a range of features to help communities organize, manage, and track documents, information, and knowledge.
    - It allows learners to quickly find and retrieve documents based on metadata, content, or keywords.

  - DocuWare:
    - DocuWare is a DMS that offers document capture, workflow automation, and secure file storage.
    - It includes collaboration tools, such as comments, annotations, and task assignments.

  - Alfresco:

- ○ It facilitates collaborations among learners with features, such as document sharing, version control, and real-time editing.
- ○ It includes collaboration tools, like comments, annotations, and task assignments, to facilitate communication and interaction with documents.

- AI-Based Learning Platforms: they leverage artificial intelligence technologies to enhance the educational experience by providing personalized, adaptive, and interactive learning opportunities. These platforms use machine learning algorithms to analyze learners' behaviors, preferences, and performances, tailoring the content to learner needs. AI-based learning platforms can be utilized in scenarios such as intelligent tutoring systems, social learning analytics, gamification for engagement, facilitating remote learning, and continuous feedback and assessment (Zhong et al. 2021) Here are some examples of AI-based learning platforms for knowledge sharing:

    - Coursera:
        - ○ It uses machine learning algorithms to recommend courses based on learner behavior.
        - ○ It employs AI for features, like the automated grading of assignments.

    - Khan Academy:
        - ○ It incorporates adaptive learning through its "Mastery Learning" approach.
        - ○ It adapts to learners' progress and provides targeted exercises.

    - Duolingo:
        - ○ Duolingo, a language learning platform, uses AI to adapt lessons based on learner performance.
        - ○ It personalizes learning paths and provides instant feedback.

    - Cognii:
        - ○ It focuses on using AI to assess and provide feedback on written responses.
        - ○ It is often used in the context of online education and assessments.

    - Squirrel AI:
        - ○ It is an adaptive learning platform that uses AI algorithms to create personalized learning paths for learners.
        - ○ It tailors the content based on learners' strengths and weaknesses.

    - SMART Sparrow:
        - ○ It is an adaptive e-learning platform that uses AI to create personalized learning experiences.
        - ○ It is commonly used in higher education and corporate training.

    - DreamBox:
        - ○ It is an AI-powered educational platform for K-8 students.
        - ○ It offers personalized math lessons based on learners' needs and progress.

    - Quillionz:
        - ○ It is an AI-based tool that helps educators generate quiz questions and assessments quickly.
        - ○ It uses natural language processing to understand and respond to user inputs.

- Open Educational Resource (OER) Platforms: provide freely accessible educational materials that can be used, adapted, and shared for teaching and learning. These platforms contribute to knowledge sharing in community learning by making educational resources available to a wide audience. OER platforms can be utilized in scenarios such as global collaboration, supporting diverse learning styles, enabling lifelong learning, enhancing inclusivity and diversity, and community-driven resource

curation (Du 2017). Here are some examples of OER platforms useful for knowledge sharing:

- OpenStax:
    - ○ It offers a library of free, peer-reviewed, openly licensed textbooks.
    - ○ The textbooks cover a variety of subjects and are suitable for college and high-school courses.
- MIT OpenCourseWare (OCW):
    - ○ It provides virtually all MIT course content for free.
    - ○ It includes lecture notes, assignments, and exams, making high-quality educational materials accessible to learners worldwide.
- OER Commons:
    - ○ It is a digital library that offers a wide range of openly licensed educational resources.
    - ○ It includes textbooks, lesson plans, videos, and more, covering various subjects and grade levels.
- MERLOT:
    - ○ It is a curated collection of free and open online teaching, learning, and faculty development services.
    - ○ It can be used for sharing teaching materials.
- Connexions:
    - ○ It is an open education platform that provides free access to high-quality educational content in a modular format.
    - ○ It covers various disciplines, and learners can create and share content.
- LibreTexts:
    - ○ It is a platform that houses a collection of open educational resources in the form of textbooks and learning materials.
    - ○ It covers a wide range of subjects.
- Wikibooks:
    - ○ It is a Wikimedia project that offers a collection of open-content textbooks.
    - ○ Learners can collaboratively create and edit content on various subjects.

The integration of such technologies to support knowledge sharing in learning communities is no longer just a trend but a fundamental shift that has redefined the landscape of education and learning. The myriad benefits derived from leveraging the above-mentioned technologies have reshaped the way knowledge is acquired and disseminated not only among community members, but also between diverse learning communities. This transformative impact underscores the necessity and significance of technology in modern education.

## 4. Discussion

Knowledge sharing within learning communities is a dynamic process crucial for fostering growth, collaboration, and innovation. Knowledge sharing is the lifeblood of learning communities, and its effectiveness is contingent on the methods and technologies employed. Effective knowledge sharing within a learning community is influenced by several crucial factors, four of which are outlined in the following paragraphs.

Knowledge sharing is inherently a 'voluntary process', where learners choose to contribute their insights, experiences, and expertise to a collective pool of knowledge. However, within any community or organizational setting, there exists a dynamic interplay between knowledge sharing and knowledge hiding. While the ideal scenario is an open and transparent exchange of knowledge, various factors influence learners' decisions to share or withhold knowledge. The dynamic nature of knowledge sharing can be influenced by factors, such as organizational culture, trust among team members, perceived value in sharing, and sometimes even competitive dynamics. Learners are more inclined to share

when they trust that their contributions is valued, and when there is a supportive and collaborative environment. Conversely, factors, like a fear of judgment, competition, or the perception of knowledge as a source of personal power, can lead to knowledge hiding. Understanding this dynamic is crucial for fostering a culture of open knowledge sharing. Thus, communities need to create an environment that encourages trust, recognizes and rewards knowledge sharing, and addresses any barriers that can lead to knowledge hiding (Pandey et al. 2021).

The 'motivational system' within a learning community plays a pivotal role in encouraging and sustaining knowledge sharing among its members. An effective system recognizes that knowledge sharing is a voluntary and dynamic process influenced by individual motivations. Incentives, both intrinsic and extrinsic, form a critical component of this motivational system. Intrinsically, learners are motivated by a sense of purpose, belonging, and personal growth. Learning communities that foster a culture of shared goals, where each learner sees the value of their contribution to the collective learning journey, are likely to see increased voluntary knowledge sharing. Recognition and acknowledgment of individual contributions, whether through formal recognition programs or informal expressions of appreciation, further enhance intrinsic motivation. Extrinsic motivators, such as rewards, career advancement opportunities, or other tangible benefits, also play a role. Learning communities can design incentive structures that tie knowledge sharing to individual and collective success. This can include awards, certifications, or opportunities for leadership roles within the community. Crucially, the motivational system needs to align with the values and objectives of the learning community. When learners perceive that their contributions are valued and that knowledge sharing is integral to the community's success, they are more likely to engage actively in sharing their expertise. Overall, a well-crafted motivational system acts as a catalyst, creating an environment where knowledge flows freely, and fostering a culture of continuous learning and collaboration (Lam and Lambermont-Ford 2010).

'Trust' is the bedrock of effective knowledge sharing within learning communities. In the intricate dance of sharing knowledge, insights, experiences, and expertise, learners need to feel confident that their contributions will be respected and used constructively. Trust is multi-faceted; it involves a belief that others have the competence to contribute meaningfully, a confidence that shared information is handled responsibly, and an assurance that credit is given where it is due. In a high-trust community, learners are more inclined to not only share what they know, but also their uncertainties and challenges, fostering a richer and more authentic exchange of knowledge. Building trust within a learning community involves transparent communication, demonstrated reliability, and a commitment to fostering an inclusive and supportive culture. When trust is nurtured, it becomes the glue that binds community members together, creating a conducive atmosphere for open dialogue, collaboration, and the collective pursuit of knowledge (Booth 2012).

The efficacy of knowledge sharing hinges significantly on the 'absorptive capacity' of learners. Absorptive capacity refers to a learner's ability to acquire, assimilate, and apply new knowledge effectively. In the context of knowledge sharing within a community, learners with higher absorptive capacity tend to grasp and integrate information more readily. Their capacity to understand and utilize shared knowledge plays a pivotal role in determining the overall success and impact of knowledge sharing initiatives. A community with members possessing robust absorptive capacity is likely to experience more meaningful and sustained knowledge exchange, fostering a dynamic environment for continuous learning and development (Chen 2004).

This discussion delved into the findings of a systematic literature review, shedding light on the importance of the diverse range of methods and technologies identified for supporting knowledge sharing within learning communities. Additionally, it provided a roadmap for enhancing knowledge sharing dynamics within learning communities and served as a guide for navigating this complex and transformative landscape. That is, through a meticulous analysis of the existing research, this systematic literature review

synthesized the current state of knowledge, pinpointing some potential methods and technologies for effective knowledge sharing within learning communities. This roadmap outlined the opportunities and challenges associated with various approaches, providing a nuanced understanding of the landscape. Researchers and educators can leverage these findings to identify the gaps in current practices and tailor interventions to address specific challenges. In essence, the results of this systematic literature review contribute helpful insights, empowering researchers and educators to make informed choices that can elevate the overall knowledge sharing experience within collaborative educational ecosystems.

The SLR performed in this study employed a rigorous and structured methodology to identify relevant articles, ensuring a comprehensive understanding of the state of the art in knowledge sharing methods within learning communities. Through stages such as automated searches, title-based selection, and reputation-based filtering, a robust selection process was established.

The findings show that the significance of methods and technologies for supporting knowledge sharing in learning communities is profound. These tool kits (methods and technologies) not only accelerate learning through knowledge sharing, but also foster collaborations and create personalized learning experiences. The identified methods and technologies for supporting knowledge sharing in learning communities form a rich tapestry. By adopting suitable or combining related methods with technologies, learning communities can create a holistic and inclusive environment, fostering a culture of continuous learning and collaborative knowledge sharing within these communities. The key lies in the adaption and use of methods and technologies that prioritize accessibility, engagement, and responsiveness to the evolving needs of learning communities and their members. Furthermore, striking a delicate balance between technological sophistication and inclusivity is crucial for maximizing the benefits of knowledge sharing initiatives within learning communities.

In learning communities, the choice of methods and technologies for knowledge sharing is contingent on various factors, for example, the nature of the content, the preferences of the learners, and the overarching goals of the educational experience. The literature review showed that varied methods, from mentorship programs to virtual workspaces, contributed to accelerated learning. Additionally, diverse methods, such as providing learning circles and discussion boards/panels, allowed for personalized learning experiences. Engaging formats, such as workshops and collaborative documentation, contributed to the application of acquired knowledge in real-world scenarios. Therefore, learners could choose the modes that aligned with their preferences, creating a tailored educational journey (Chau et al. 2003; Koskinen and Pihlanto 2008; Islam 2012; Szulczyńska and Majewska 2014). It should be added that the success of technological tools relies on user adoption and technological literacy. For example, for synchronous interactions and real-time collaborations, video conferencing tools prove invaluable, offering a face-to-face connection, even in virtual spaces. Discussion forums and collaborative platforms excel in asynchronous learning scenarios, fostering ongoing dialogues and community engagements. Document management systems are particularly effective for organizing and sharing structured knowledge, such as manuals and procedural guides. Blogs and microblogging platforms are well-suited for quick updates, reflections, and sharing insights in a more informal setting. E-learning platforms, including MOOCs and LMSs, provide structured courses and assessments for comprehensive learning experiences. LMSs streamline educational processes, enabling learners to access, absorb, and apply knowledge efficiently. For practical skills development, virtual and augmented reality technologies offer immersive, hands-on learning opportunities. Social media platforms contribute to community building and informal knowledge exchange. Social media platforms and community forums enable professionals to network, fostering relationships that go beyond knowledge sharing. These connections can lead to mentorship, collaboration on projects, and job opportunities. Podcasts cater to auditory learners and those on the go, offering insights into and discussions on diverse topics. The use of technologies raises concerns about data security and privacy. Hence,

safeguarding sensitive information becomes paramount, necessitating robust measures and policies (Aggarwal et al. 2011; Aggarwal et al. 2011; Tiwari 2022; Tiwari 2022).

Furthermore, learning communities thrive on the exchange of knowledge, collaboration, and shared experiences. The integration of a diverse range of methods and technologies plays a pivotal role in fostering a dynamic and enriching learning environment. The following examples show how these elements contribute to enhancing knowledge sharing within learning communities.

- Learners have diverse learning preferences and styles. Some prefer visual learning through videos, while others excel in interactive workshops. By offering a variety of methods—such as video lectures, written documentation, live workshops, and collaborative projects—learning communities can accommodate different learning styles, making education more inclusive (Efrati et al. 2014).
- Accessibility is crucial for effective learning, considering the varied needs of learners, including those with disabilities. Technologies, like closed captions in videos, screen readers for written content, and accessible virtual reality experiences, ensure that learning resources are inclusive and can be accessed by a broader audience (Zhang et al. 2020).
- Active participation enhances engagement and knowledge retention. Discussion forums, collaborative platforms, and social media integration within learning communities encourage learners to actively contribute, share insights, and engage in meaningful dialogues, creating a vibrant and participative community (Tseng and Kuo 2014).
- Modern learners often seek flexibility and convenience in their education journey. Online learning platforms, webinars, and asynchronous communication methods provide the flexibility for learners to access content at their own pace and convenience, catering to the demands of varied schedules and commitments (Pang et al. 2020; Zafar Yaqub and Alsabban 2023).
- Collaborative learning promotes the exchange of diverse perspectives and the co-creation of knowledge. Technologies, like collaboration platforms, wikis, and group projects, facilitate collaborative learning, allowing learners to work together, share knowledge, and collectively contribute to the knowledge pool (Aggarwal et al. 2011).
- Gamification introduces elements of play and competition, making learning enjoyable. Incorporating gamified elements into educational content encourages friendly competition, motivates learners to achieve milestones, and adds an element of excitement to the learning process (Friedrich et al. 2020).
- Learning communities are often diverse, with members from various cultural backgrounds. Virtual communication tools, video conferencing, and social media enable global interaction, allowing learners to gain insights from diverse perspectives, and fostering a rich and multicultural learning experience (Aljuwaiber 2019).
- Multimedia content enhances engagement and aids comprehension. Podcasts, videos, and interactive simulations provide a multi-sensory learning experience, catering to auditory and visual learners and making complex concepts more accessible (Louadi et al. 2023).
- LMS platforms provide structured and organized learning experiences. LMSs offer centralized platforms for course content, assessments, and collaborative activities, providing a structured and streamlined approach to learning (Cuéllar et al. 2011).

Despite the positive advantages of methods and technologies, some researchers (Babu and Gopalakrishnan 2008) argue that implementing KM programs does not necessarily require technology, as KM is basically about people, not technologies. To some extent they are right. However, it is unequivocally impractical to facilitate effective knowledge sharing within a community—be it small or large and geographically dispersed—without the integration of technology. For example, the support of Information Technology (IT) can be categorized into two crucial approaches: (a) the utilization of a suitable repository for storing and disseminating knowledge and (b) the application of communication mediums

to facilitate the exchange and transportation of knowledge among individuals. The first approach is the use of a proper repository or repository model of a KM system, which is related to database management and organizational memory. The IT is used to capture, categorize, search, subscribe to pertinent content or knowledge, and present it in more meaningful formats that transcend various usage contexts. Additionally, IT plays a pivotal role in transforming tacit knowledge into an explicit form. The second approach revolves around leveraging a communication medium, akin to the network model within a KM system—an extension of various computer-mediated electronic communication methods. This facet is instrumental in facilitating interactions, direct communication, and contact among individuals.

The findings reveal that each method and technology has its unique strengths and challenges. The selection of methods and technologies for knowledge sharing in learning communities depends on a multitude of factors, including, but not limited to, contextual characteristics; learning objectives; learner preferences; technological infrastructure, availability, and capability; real-time interaction requirements; self-paced learning needs; nature of content; community engagement; and feedback mechanisms. In essence, the effectiveness of knowledge sharing methods and technologies is a nuanced interplay of these factors, and thoughtful consideration of each is necessary for designing successful learning experiences within communities. A tailored selection and integration of these methods and technologies will align with the unique requirements and dynamics of different learning community scenarios, enhancing the overall effectiveness of knowledge sharing. The integration of a diverse range of methods and technologies in learning communities contributes to a holistic and inclusive educational experience. By recognizing and leveraging the strengths of each method and technology, learning communities can create an environment that not only shares knowledge effectively, but also nurtures a culture of continuous learning and collaboration.

Considering the findings of this work, it can be concluded that the first and second hypotheses are positively validated to a reasonable extent.

### 4.1. Challenges and Considerations

Implementing methods and technologies for knowledge sharing within learning communities comes with various challenges and considerations. Here are some key aspects to consider (Widen-Wulff 2007; Dube and Ngulube 2012; Asrar-ul-Haq and Anwar 2016):

- Diversity of Learning Styles:
    - Challenge: learning communities often comprise learners with diverse learning preferences and styles.
    - Consideration: implementing a mix of methods and technologies (e.g., visual content and interactive discussions) to cater to diverse learning styles.
- Technical Infrastructure:
    - Challenge: variability in the technical infrastructure and digital literacy levels among community members.
    - Consideration: ensure that chosen technologies are accessible and user-friendly and provide training or support for those less familiar with digital tools.
- Privacy and Security:
    - Challenge: balancing the need for open knowledge sharing with concerns about privacy and data security.
    - Consideration: employ secure platforms, clearly communicate privacy policies, and establish guidelines for responsible knowledge sharing.
- Community Engagement:
    - Challenge: sustaining active participation and engagement over time.

- ○ Consideration: implement gamification, discussion forums, and other interactive features to maintain interest, and regularly assess and respond to the community's evolving needs.
- Scalability:
  - ○ Challenge: adapting methods and technologies to accommodate the growth of the learning community.
  - ○ Consideration: choose scalable technologies that can handle increased learner numbers and evolving needs without compromising learner experience.
- Knowledge Relevance:
  - ○ Challenge: ensuring that shared knowledge remains relevant and aligns with the community's objectives.
  - ○ Consideration: establish mechanisms for knowledge curation, learner feedback, and regular updates to keep knowledge current and valuable.
- Digital Inclusion:
  - ○ Challenge: addressing issues of accessibility and inclusivity for learners with varying levels of technological access.
  - ○ Consideration: strive for inclusivity by providing alternatives for those with limited digital access, such as offline resources or community meet-ups.
- Training and Support:
  - ○ Challenge: assisting learners in becoming proficient with new tools and technologies.
  - ○ Consideration: offer training sessions, tutorials, and ongoing support to ensure that community members feel confident and competent in using the provided tools.
- Cultural Sensitivity:
  - ○ Challenge: navigating cultural differences that can impact the acceptance and effectiveness of certain methods or technologies.
  - ○ Consideration: conduct cultural assessments, seek feedback, and be flexible in adapting methods to align with cultural norms and preferences.
- Evaluation and Feedback:
  - ○ Challenge: measuring the effectiveness of knowledge sharing methods and technologies.
  - ○ Consideration: establish clear metrics for success, regularly gather feedback from community members, and be willing to adapt based on evaluation results.

Addressing these challenges and considering these factors contribute to the successful implementation of methods and technologies for knowledge sharing within learning communities.

### 4.2. Future Work

Exploring the future avenues for methods and technologies for supporting knowledge sharing in learning communities involves several exciting possibilities. Here are potential areas for future work:

- Integration of AI: investigate how AI technologies can be integrated into learning communities to enhance knowledge sharing. This can involve personalized learning recommendations, intelligent chatbots for instant assistance, and predictive analytics to identify and address knowledge gaps.
- Blockchain for Credentialing: research the application of blockchain technology for secure credentialing within learning communities. This can provide a transparent and tamper-proof record of achievements and skills, fostering trust and recognition.

- Enhanced Collaboration Platforms: develop advanced collaborative platforms that go beyond traditional tools. This can involve real-time co-authoring, interactive project management, and AI-driven features that facilitate seamless collaboration.
- Social Media Analysis: explore advanced analytics and sentiment analysis on social media platforms dedicated to learning communities. Understand the trends, preferences, and sentiments to tailor knowledge sharing strategies effectively.
- Longitudinal Studies on Impact: conduct longitudinal studies to assess the long-term impact of implemented technologies and methods. Understand how these interventions influence the learning outcomes, collaboration dynamics, and overall community engagement over time.

By delving into such future directions, researchers and practitioners can contribute to the continuous improvement of methods and technologies for supporting knowledge sharing in learning communities, ensuring they remain effective, ethical, and aligned with the evolving needs of learners.

**Author Contributions:** Conceptualization, M.Z.; methodology, M.Z.; software, A.E.; validation, M.Z. and A.E.; formal analysis, M.Z. and A.E.; investigation, M.Z.; resources, M.Z. and A.E.; data curation, M.Z. and A.E.; writing—original draft preparation, M.Z.; writing—review and editing, M.Z. and A.E.; visualization, M.Z. and A.E.; supervision, M.Z.; project administration, M.Z. All authors have read and agreed to the published version of the manuscript.

**Funding:** This research received no external funding.

**Institutional Review Board Statement:** Not applicable.

**Informed Consent Statement:** Not applicable.

**Data Availability Statement:** No new data were created or analyzed in this study. Data sharing is not applicable to this article.

**Acknowledgments:** The authors are grateful to the anonymous referees for their valuable suggestions.

**Conflicts of Interest:** The authors declare no conflicts of interest.

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
