# Peer review of "Methods and Technologies for Supporting Knowledge Sharing within Learning Communities: A Systematic Literature Review"

_admsci, doi:10.3390/admsci14010017_

Round 1

Reviewer 1 Report

Comments and Suggestions for Authors

1.The topic is interesting and the review could be useful.

2.The authors must underline the fact that knowledge sharing is a voluntary process and there is always a dynamics between knowledge sharing and knowledge hiding. The discussion on this dynamics is completely missing.

3.Knowledge sharing depends on the motivation of the individual who decides to share some of his knowledge. Thus, the motivational system of the learning community is extremely important. The authors should discuss this issue.

4.There are some paragraphs where the authors consider the concepts of "information" and "knowledge" being similar. They are not similar. We strongly recommend the authors to read at least the following paper that explains this issue: Bratianu, C. & Bejinaru, R. (2023). From knowledge to wisdom: Looking beyond the knowledge hierarchy. Knowledge, 3(2), 196-214. https://doi.org/10.3390/knowledge3020014.

5.Knowledge sharing is based on trust. however, there is no discussion concerning trust. The authors must introduce in their findings th erole of trust in supporting knowledge sharing. Otherwise, the paper looks like an inventory of programs and platforms which may help people in knowledge sharing.

6.The result of knowledge sharing depends on the absorptive capacity of learners. The authors should discuss on the role of absorptive capacity.

Reviewer 2 Report

Comments and Suggestions for Authors

The given paper provides a systematic literature review covering the two areas of education (learning communities) and knowledge management (knowledge sharing) and how knowledge sharing may be applied within learning communities.

While it is an instersting topic to consider, both areas are not new and KM itself talks about using communities for knowledge development and sharing, therefore there are some points to criticise about your research questions and the submitted version of the article.

1. Introduction:

While the first section looks sound and comprehensible regarding the characteristics of learning communities and knowledge sharing it it not entirely clear to the reader why you are doing this systematic literature review and where is your motivation to focus on knowledge sharing within learning communities? Is there a background issue where you need to find out the best way of knowledge sharing or criteria of the size and type of a learning community and the best ways to share knowledge depending on the type of community?
In lines 34 - 37 different types of learning communities are mentioned, but later on it is not reflected which ones would benefit best regarding which knowledge sharing methods. Based on lines 311-313 I understand that the review is done from the perspective of education and community learning not from the perspective of knowledge management, as this literature already provides methods and KMS to support knowledge sharing. Communities of practice or collaboration networks are one method of knowledge sharing and creation in KM. Perhaps adding a bit more KM literature would benefit the work.  

Knowledge sharing is not introduced as part of knowledge management in section 1.2. In KM there are models which also cover knowledge distribution. This term or how it is the same or different to knowledge sharing is not mentioned. How about knowledge distribution principles such as pull and push-principle to request or provide knowledge within a community?

In some paragraphs the literature references are to sparse, so the statements can not be related accordingly, e.g. lines 196-198 where you talk about vibrant hubs without a reference: "Knowledge sharing is the lifeblood of learning communities. It transforms these spaces from passive environments into vibrant hubs where ideas flow freely, and learners actively contribute to the collective pool of understanding."

2. Systematic Literature Review
Personally, I find it a bit strange that there is only one rather generic research question to your work and that your hypothesis seems straightforward that knowledge sharing within learning communities can be supported if potential and effective methods and technologies are applied. You already detail this in the introduction as a somehow given situation, therefore I would have found it more interesting for the reader to see which methods and technologies for knowledge sharing are best used for which scenarios or types in learning commuinities.

Regarding the search terms the question remains if you might have gotten additional results if you considered knowledge distribution as a secondary search term. Knowledge development or creation within a community might also be an additional search term, but would have a slightly different perspective. If you used them the question would be more focussed on the aspect of how learning commuinities would be able to create new knowledge within them.

3. Identified methods and technologies

While this reads quite comprehensible and details methods and technologies I would recommend to rework the listings and add fitting references to each bullet point. In line 426 there are three references mentioned but it is unclear for the reader if they are the basis for all following bullet points from lines 429-522. As it is a systematic literature review there should be more references mentioned which provide the foundation for each method. The same could be said for the technologies listed starting from 542. As each technology is also defined, there should be a reference to each definition, e.g. lines 543-545 defining a LMS and its characteristics but not naming a reference.

Another point which should be improved is that it is not realy clear how you gathered and selected the listed tools of each technology, e.g. Moodle, Canvas etc. Are they the most talked about in your literature review or searched parallel based on websites giving statistics about their current use? Some literature in your survey is already a few years old, so it provides the question if the tools are still up-to-date or how you made sure to present the current status of the tools in use? This comment applies to the lines 550 to 996. You have essentially several pages of tools without a single reference from your literature review, which definitely should be improved.

4. Discussion
In line 1011 you talk about the provision of a roadmap for enhancing knowledge-sharing dynamics. I did not detect this in the following paragraphs, or do you mean section 4.2 about future work?
For me it would have been interesting to detect which methods and tools are already established in education and which new ones you identified from KM for knowledge sharing.

As mentioned before, another intersting aspect would have been to see which methods and tools would be most fitting for which type of learning community or size of the community. You start this a bit in line 1050, but there are again not references founding your statements.

In 1093 you conclude that your hypothesis is fulfilled, but as mentioned in 2., this has been somehow a bit of a given based on your introduction.

It is good that you give a section about challenges and considerations, but adding some references would be beneficial here as well. It could be more extensive as well.

References:

In your references I am unable to detect some works about knowledge management.

Some other systematic literature reviews also provide a table detailing the used references and criteria to map the references to the research questions or hypotheses.

---

To conclude on a positive point, you detected and mentioned the main methods and technologies of KM and KMS in your survey in section 3, but may of those could also be detected when only focussing on KM literature.

Comments on the Quality of English Language

The paper is easily readable, there are only some minor typos.

Round 2

Reviewer 1 Report

Comments and Suggestions for Authors

The authors answered all recommendations made by the reviewers.

Author Response

Dear Reviewer,

We appreciate the time and effort that you dedicated to providing feedback on our manuscript and are grateful for the insightful comments and valuable improvements to our paper. 

Reviewer 2 Report

Comments and Suggestions for Authors

Thank you for adapting your work based on the previous reviews. I think it is improved and considers the KM aspects more, also adding references and a second hyotheses to your work. Also adding which method may be used for wich scenario is quite helpful.

Regarding 2.1 (lines 437-438), you seem to have added additional search terms for your survey, but the number of your results (N=198) stays the same as in the version before, or did I missunderstand the mentioned lines? Usually if you add more search terms the results would differ. Perhaps you modify theses lines again and stay with the old search terms, but mention that you also consider these new ones for your analysis of the deteced references?

As you added a number of new references compared to the previous version, I assume those are the ones which were part of your search results and which are now referenced in the sections more explicitly, compared to the version before.

Author Response

Dear Reviewer, 

We appreciate the time and effort that you dedicated to providing feedback on our manuscript and are grateful for the insightful comments and valuable improvements to our paper. 

Considering your pretty good remark in the second review, we have made the related and needed changes in lines 443, 446, and 448, and also in Figure 1 and Figure 2.